# Infants' looking preferences for social versus non-social objects reflect genetic variation

Ana Maria Portugal [1,2] ✉, Charlotte Viktorsson [1], Mark J. Taylor[3], Luke Mason[4], Kristiina Tammimies [2,5], Angelica Ronald [6] & Terje Falck-Ytter [1,2,7] ✉

To what extent do individual differences in infants' early preference for faces versus non-facial objects reflect genetic and environmental factors? Here in a sample of 536 5-month-old same-sex twins, we assessed attention to faces using eye tracking in two ways: initial orienting to faces at the start of the trial (thought to reflect subcortical processing) and sustained face preference throughout the trial (thought to reflect emerging attention control). Twin model fitting suggested an influence of genetic and unique environmental effects, but there was no evidence for an effect of shared environment. The heritability of face orienting and preference were 0.19 (95% confidence interval (CI) 0.04 to 0.33) and 0.46 (95% CI 0.33 to 0.57), respectively. Face preference was associated positively with later parent-reported verbal competence ($\beta = 0.14$, 95% CI 0.03 to 0.25, $P = 0.014$, $R^2 = 0.018$, $N = 420$). This study suggests that individual differences in young infants' selection of perceptual input—social versus non-social—are heritable, providing a developmental perspective on gene–environment interplay occurring at the level of eye movements.

From looking and interacting with other people, infants get experiences that contribute to shaping social brain circuits and social cognition. At the same time, the developing infant is confronted with the massive task of learning about non-social objects and events. Whether an infant looks at faces or at non-social objects at any moment in time can reflect both bottom-up and top-down processes, including interests, understanding and motivation, and the maturation of the cognitive system[1,2]. Atypical attention to social versus non-social objects has been implicated in autism, a heritable neurodevelopmental condition partly defined by social communication difficulties[3–6]. However, also in the typical population, there is substantial variation with regard to social versus non-social visual preferences[5,7,8], and recent data suggest that specific aspects of social preferences such as attention to eyes versus mouth of other people's faces are highly heritable in

infants[9] and young children[10]. The current study evaluated the extent to which visual preference for faces versus non-social information in early infancy reflects genetic variation in the population (which, at the extreme, could be associated with heritable clinical conditions such as autism[5,11,12]).

Faces selectively attract attention already at birth and act as a catalyst for cognitive, social and emotional development[13–16]. An early bias to orient to faces (fast first looks at face-like configurations in the periphery) is proposed to be subcortically mediated and present at birth, whereas sustained looking at faces requires later maturing cortical top-down structures[15]. By 6 months of age, top-down control enables flexible looking behaviour permitting the infant to preferentially attend to the face but also to shift attention away from it and to the other stimuli in the environment[15,17,18].

[1]Development and Neurodiversity Lab (DIVE), Department of Psychology, Uppsala University, Uppsala, Sweden. [2]Center of Neurodevelopmental Disorders (KIND), Centre for Psychiatry Research, Department of Women's and Childrn's Health, Karolinska Institutet & Stockholm Health Care Services, Stockholm, Sweden. [3]Department of Medical Epidemiology and Biostatistics, Karolinska Institutet, Stockholm, Sweden. [4]Department of Forensic and Neurodevelopmental Sciences, Institute of Psychiatry, Psychology and Neuroscience, King's College London, London, UK. [5]Astrid Lindgren Children's Hospital, Karolinska University Hospital, Stockholm, Sweden. [6]School of Psychology, Faculty of Health and Medical Sciences, University of Surrey, Guildford, UK. [7]Swedish Collegium for Advanced Study, Uppsala, Sweden. ✉e-mail: ana-maria.portugal@psyk.uu.se; terje.falck-ytter@psyk.uu.se

**Table 1 | Descriptive statistics of the primary face looking measures. Statistics presented as mean (s.d.)/min–max**

| | Overall | MZ females | MZ males | DZ females | DZ males | Skewness |
|---|---|---|---|---|---|---|
| **N** | 536 | 135 | 158 | 116 | 127 | |
| **Age (in days)** | 168 (9) 145–203 | 168 (9) 153–194 | 167 (8) 150–187 | 167 (8) 153–189 | 168 (10) 145–203 | 0.62 |
| **No. valid trials** | 5.77 (0.52) 4–7 | 5.76 (0.52) 4–7 | 5.75 (0.55) 4–6 | 5.75 (0.56) 4–6 | 5.82 (0.44) 4–6 | −2.12 |
| **Proportion missing gaze samples** | 0.28 (0.13) 0–0.61 | 0.28 (0.13) 0.01–0.61 | 0.28 (0.12) 0.05–0.58 | 0.29 (0.13) 0.02–0.60 | 0.27 (0.14) 0–0.58 | 0.13 |
| **Face orienting (Proportion first look at the face)[*]** | 0.30 (0.19) 0–1 | 0.29 (0.19) 0–0.80 | 0.27 (0.2) 0–0.83 | 0.34 (0.19) 0–0.83 | 0.30 (0.19) 0–1 | 0.51 |
| **Face preference (Proportion on face)[*]** | 0.44 (0.14) 0.09–0.81 | 0.45 (0.14) 0.11–0.78 | 0.42 (0.15) 0.09–0.80 | 0.45 (0.13) 0.11–0.70 | 0.43 (0.14) 0.15–0.81 | 0.14 |
| **Efficiency of visual exploration (No. objects explored, maximum 5 during 10 s)** | 3.64 (0.54) 2–4.83 | 3.61 (0.54) 2.17–4.83 | 3.66 (0.52) 2.17–4.83 | 3.60 (0.56) 2–4.83 | 3.69 (0.55) 2.25–4.67 | −0.34 |

[*]Chance level of face looking was 0.20 (1 in 5 objects was a face). MZ, monozygotic; DZ, dizygotic.

**Table 2 | Descriptive statistics of parent-reported development measures at 14 and 24 months. Statistics presented as mean (s.d.)/min–max**

| | N (n girls) | Age (in days) | Score (both sexes) | Females' score | Males' score | Skewness |
|---|---|---|---|---|---|---|
| **ECBQ self-regulation** | | | | | | |
| At 14 months | 436 (196) | 442 (20) 386–525 | 4.32 (0.58) 2.40–6.15 | 4.30 (0.59) 2.40–6.15 | 4.33 (0.56) 2.40–5.76 | 0.10 |
| At 24 months | 358 (180) | 755 (26) 707–920 | 4.56 (0.65) 2.73–6.25 | 4.69 (0.61) 2.73–6.00 | 4.44 (0.66) 2.80–6.25 | −0.17 |
| **ITC social communication** | | | | | | |
| At 14 months | 418 (196) | 444 (20) 387–525 | 34.93 (6.85) 11–51 | 35.84 (6.09) 20–51 | 34.13 (7.38) 11–51 | −0.40 |
| **CDI vocabulary** | | | | | | |
| At 14 months | 420 (197) | 443 (20) 386–502 | 82.52 (64.47) 1–332 | 92.90 (61.85) 1–305 | 73.35 (65.48) 3–332 | 1.16 |
| At 24 months | 335 (168) | 756 (23) 707–920 | 207.62 (156.91) 1–689 | 251.66 (143.71) 18–627 | 163.32 (157.55) 1–689 | 0.72 |

ECBQ, Early Childhood Behavior Questionnaire; ITC, Infant Toddler Checklist; CDI, Communicative Development Inventory.

In this Article, based on a pre-registered analysis plan[19], we studied the genetic and environmental influences underlying individual differences in two early emerging aspects of selective attention to social versus non-social information: orienting (looking first at faces rather than non-social objects), and sustained preference (ratio of looking time in the face relative to other objects). We also assessed the infants' efficiency of visual exploration, defined as how many of the objects in the stimulus array (social and non-social) infants looked at during the first 10 s following stimulus onset. We expected that variation in all three phenotypes studied would have a significant genetic component[9,10,20]. We also assessed the aetiological link between the eye-tracking phenotypes, but due to lack of previous research we had no specific hypotheses. Next, given the links reported between visual attention to faces and autism[3–6] and between attention control and attention-deficit/hyperactivity disorder (ADHD)[11], we also tested whether the different emerging aspects of attention to faces early in life were associated with polygenic scores (common genetic variance) and later traits related to autism and ADHD; specifically, we expected face preference at 5 months to be related to higher social communication abilities and the efficiency of visual exploration to be related to later self-regulation. Finally, because greater attention to faces in infancy is thought to predict better language outcomes later in life (for example, refs. 21,22), we also studied whether the different aspects of attention to faces were

associated with later language skills, specifically we expected that face preference at 5 months would be related to higher language abilities. Data came from the Babytwins Study Sweden (BATSS)[23], a Swedish community sample of dizygotic and monozygotic 5-month-old twins who went through gaze-based experimental measurements of looking at faces presented together with other non-face objects in a five-item array.

We used a classical twin modelling approach, in which one compares the level of within-pair similarity separately for monozygotic twins (MZ; who share 100% of their segregating genetic material) and dizygotic twins (DZ; who on average share 50%). Univariate twin models estimate the relative contribution of genetic and environmental factors to the variation in a phenotype, by comparing the correlation between twins; while bivariate twin models further estimate the relative contributions of genetic and environmental factors to the covariation between two phenotypes, by comparing cross-trait cross-twin (CTCT) correlations, that is, the correlation between one phenotype for one twin and another phenotype for their co-twin. The variation or covariation can be decomposed into additive genetic influences (A; heritability, which increases twin similarity), non-shared environment (E; environmental influences that differ between twins and decrease twin similarity, including measurement error), and shared environment (C; environmental influences that increase twin similarity regardless of zygosity, for example, family socioeconomic status).

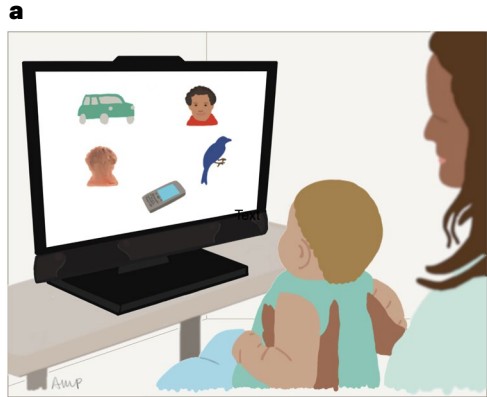

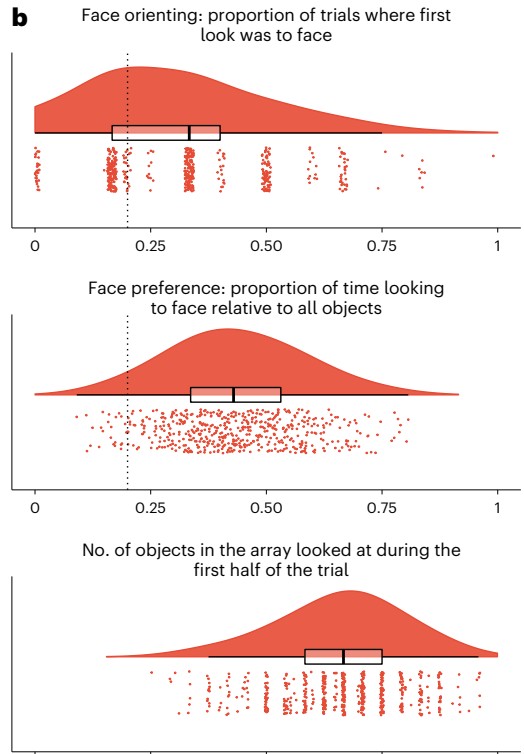

**Fig. 1 | The face pop-out paradigm—illustrative set-up and primary looking measures data plots. a**, An infant viewing one trial of the face pop-out task. Illustration by author A. M. P. **b**, Raincloud plots[54] (centre lines represent the median, box limits represent upper and lower quartile, whiskers represent 1.5× interquartile range, and outliers are not presented) of the three primary looking measures derived from the task, across 536 5-month-old infants: face orienting (mean was significantly above chance level, highlighted as a dashed vertical line;

one-sample two-tailed $V_{Twin 1}(273) = 25{,}558$, $P < 0.001$, $d = 0.54$, 95% CI 0.25 to 0.33; $V_{Twin 2}(261) = 23{,}159$, $P < 0.001$, $d = 0.49$, 95% CI 0.25 to 0.33), face preference (mean was significantly above chance level, highlighted as a dashed vertical line, $t_{Twin 1}(273) = 28.29$, $P < 0.001$, $d = 1.71$, 95% CI 0.42 to 0.46; $t_{Twin 2}(261) = 26.31$, $P < 0.001$, $d = 1.63$, 95% CI 0.41 to 0.45), and efficiency of visual exploration (number of objects explored during the first 10 s of trial).

### Table 3 | Twin correlation coefficients (95% CIs are shown in brackets) for the primary face looking measures, separate for MZ and DZ pairs

| | N (twin pairs[*]) | Face orienting (proportion first look at the face) | Preference for face (proportion on face) | No. of objects explored (in 0–10s) |
|---|---|---|---|---|
| **MZ** | 155 | 0.20 [0.03 to 0.34] | 0.46 [0.32 to 0.57] | 0.05 [−0.12 to 0.22] |
| **DZ** | 130 | 0.05 [−0.13 to 0.23] | 0.21 [0.01 to 0.38] | 0.15 [−0.03 to 0.31] |

[*]Incomplete twin pairs Correlations were derived from the univariate twin models where means and variances were equated across twin order and zygosity, and age and sex were included as covariates.

## Results

Sample descriptive statistics are presented in Tables 1 and 2. As expected, we found a face bias significantly above chance level, for both face orienting (proportion of trials with first look at the face; one-sample two-tailed $V_{Twin 1}(273) = 25{,}558$, $P < 0.001$, $d = 0.54$, 95% CI 0.25 to 0.33; $V_{Twin 2}(261) = 23{,}159$, $P < 0.001$, $d = 0.49$, 95% CI 0.25 to 0.33) and preference (proportion of time spent looking at the face; $t_{Twin 1}(273) = 28.29$, $P < 0.001$, $d = 1.71$, 95% CI 0.42 to 0.46; $t_{Twin 2}(261) = 26.31$, $P < 0.001$, $d = 1.63$, 95% CI 0.41 to 0.45; Fig. 1). The univariate twin correlations for the three measures are presented in Table 3. Twin modelling assumptions (of equality of mean and variances across twin order and zygosity) were met for the three measures (Supplementary Tables 1, 3 and 5). For the efficiency of visual exploration (social and non-social objects), a univariate twin modelling analysis indicated no genetic influences (reported in

Supplementary Table 2; for general information about the twin model fitting approach, also see Methods). Therefore, polygenic scores analysis involving efficiency of visual exploration were not conducted. Further, to simplify the multivariate twin analysis, we chose to only include the two phenotypes with genetic effects (univariate twin modelling analysis reported in Supplementary Tables 4 and 6).

Twin modelling assumptions for the bivariate twin analysis were met (equality of phenotypic and CTCT correlations across twin order and zygosity; Supplementary Table 7). The phenotypic correlation between face orienting and face preference was positive and moderate ($r_{Ph} = 0.30$, 95% CI 0.22 to 0.37, $\Delta\chi^2$(Δd.f. 1) of 46.58, $P < 0.001$). A Cholesky bivariate twin model was used to examine genetic influences on face preference that were either unique to face preference or shared with face orienting (age and sex included as covariates). The AE model, that is, the model with additive genetic influences (A) and non-shared environment (E) and without shared environment influences (C), was selected and reported in Table 4 and Fig. 2 on the basis that it was the non-significant model (that is, did not have a significantly poorer fit compared with the ACE model, that is, the model with A, C, and E influences) with the lowest Akaike information criterion (AIC) value (for completeness, full ACE estimates are reported in Supplementary Table 8). The heritability of face orienting was 0.19 (95% CI 0.04 to 0.33), and the heritability of face preference was 0.46 (95% CI 0.33 to 0.57). The bivariate results showed that 97% of total E influencing face preference was unique to that variable and not shared with face orienting. Of the total genetic influences on face preference (A = 0.46, as above), 0.16 (95% CI 0.03 to 0.51) were shared with, and 0.29 (95% CI 0 to 0.45) were unique from, orienting to faces (Fig. 2). A follow-up analysis testing

**Table 4 | Bivariate twin model fit statistics for face orienting and face preference**

| Model | No. of parameters | −2LL | d.f. | AIC | Comparison model | Δχ² | Δd.f. | P value |
|---|---|---|---|---|---|---|---|---|
| Fully sat. | 32 | 1,151.37 | 1,040 | −928.63 | NA | NA | NA | NA |
| **ACE** | | | | | | | | |
| ACE | 15 | 1,167.82 | 1,057 | −946.18 | Fully sat. | 16.45 | 17 | 0.492 |
| **ACE-nested models** | | | | | | | | |
| **AE** | **12** | **1,169.33** | **1,060** | **−950.67** | **ACE** | **1.51** | **3** | **0.680** |
| CE | 12 | 1,175.34 | 1,060 | −944.66 | ACE | 7.52 | 3 | 0.057 |
| E | 9 | 1,211.82 | 1,063 | −914.18 | ACE | 44.00 | 6 | <0.001 |
| **AE-nested models** | | | | | | | | |
| Unique path of 0 | 11 | 1,172.28 | 1,061 | −949.72 | AE | 2.95 | 1 | 0.086 |
| Shared path of 0 | 11 | 1,179.41 | 1,061 | −942.59 | AE | 10.08 | 1 | 0.001 |

The best-fitting model was selected on the basis of non-significance (meaning that there was no decrement in fit compared with the saturated or the genetic model, indexed by the $\chi^2$ distribution) and the AIC fit statistic (which incorporates information about both explained variance and parsimoniousness). The fully sat. model is the fully saturated model of the observed data, which models the means and variances for both variables, and the phenotypic and CTCT correlations between the two variables, separately for each twin in a pair and across zygosity. In bold: the best-fitting model was non-significant with the lowest AIC. −2LL, fit statistic, which is minus two times the log-likelihood of the data. d.f., degrees of freedom. AIC, fit statistic—lower values denote better model fits. $\Delta\chi^2$, difference in −2LL statistic between two models, distributed $\chi^2$. $\Delta$d.f., difference in degrees of freedom between two models.

two nested models constraining the shared or the unique influences confirmed there was evidence for significant shared genetic influences between the two phenotypes and no evidence for significant unique genetic variance on face preference (Table 4).

### Association between face and eye-versus-mouth preferences

Given that infants' preference for eyes (rather than mouth) when looking at faces has been found to correlate with face preference[24] and has a substantial heritability ($h^2 = 0.57$)[9], we investigated the link between face preference (this study) and the previously analysed social looking phenotype. While there was a small but significant positive phenotypic association ($r_{Ph} = 0.11$, 95% CI 0.01 to 0.20, $\Delta\chi^2(\Delta$d.f. 1$) = 5.05$, $P = 0.025$) between these phenotypes, there was no evidence for a genetic correlation ($r_A = 0.10$, 95% CI −0.13 to 0.31), and independent genetic factors contributed to eye (versus mouth) preference and face (versus object) preference (for full results, see Supplementary Result 1; this post hoc analysis was not pre-registered).

### Associations with polygenic scores

We found no evidence for associations between the face looking measures (face orienting and face preference) and polygenic scores for autism, ADHD, schizophrenia, depression and bipolar disorder (Supplementary Table 9).

### Longitudinal phenotypic associations

We found no evidence for associations between the looking measures and subsequent parent-reported measures of language, socio-communication and self-regulation (Table 5), with the exception of a statistically significant positive association between preference for the face at 5 months and receptive vocabulary (comprehension in the Communicative Development Inventory (CDI)) at 14 months ($\beta = 0.14$; 95% CI 0.03 to 0.25, $P = 0.014$; $R^2 = 0.018$, $N = 420$).

### Discussion

Attention preferences for social information, and looking at faces more specifically, have been suggested to play a key role in the development of social cognition[25]. Against this background, it is striking that we found substantial variability in infants' attention to faces at an early point in life—before brain systems supporting social communication are fully developed (Fig. 1). As predicted, both face orienting at the beginning of the task and sustained face preference were heritable phenotypes. In contrast, we found no evidence for shared environment or biological sex effects in infants' tendency to preferentially attend to

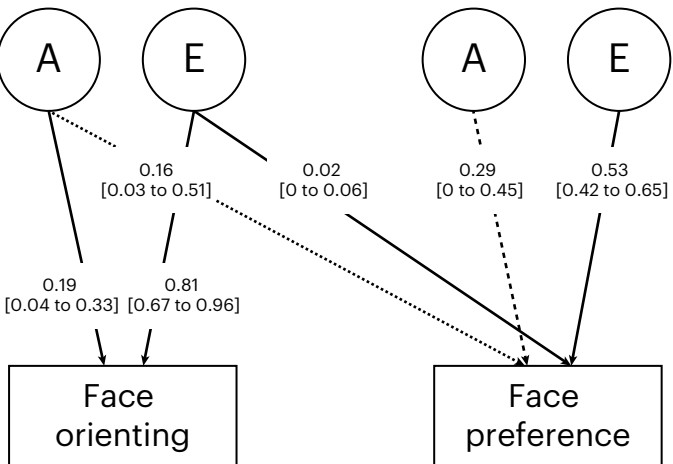

**Fig. 2 | Schematic AE bivariate twin model for face orienting and face preference.** Twin structural equation model-fitting was used to decompose the variance in face orienting and face preference into genetic (A) and unique environment (E) influences. Point estimates are shown with 95% CIs in brackets. The dotted line represents the shared genetic influences (significant), whereas the dashed line represents the unique genetic influences (non-significant).

faces versus non-social objects. The pattern of these results supports the view that, already during infancy, there is genetic variability to curate one's visual environment via looking behaviours[9,10,20], and that this applies to such broad categories as social versus non-social stimuli. This can be seen as a type of gene–environment correlation[26] appearing very early in life. Active exposure to different visual environments entails different learning opportunities (active gene–environment correlation) and, because our point of gaze is visible to others, may evoke different reactions from other people (evocative gene–environment correlation). Visual environment selection by means of selective attention is one of the first ways infants can actively create or constrain their own visual experience and social interactions, emerging before other exploratory behaviours such as pointing, grasping or crawling to targets or partners.

Face preference, indexed by the looking time to the face relative to looking time to all objects in the scene, had a heritability of 46%, while face orienting, indexed by the proportion of first looks at the face, had a heritability of only 19%. A similar result was obtained when we used

**Table 5 | Results of multiple GEEs analyses with 14 months and 24 months parent-reported measures as outcome variables, the age, sex, and looking measures measured at 5 months as predictors, and twin pair identification as cluster-defining variable**

| ECBQ self-regulation at 14 months | β | s.e. | 95% CI | Uncorrected P value | FDR threshold |
|---|---|---|---|---|---|
| Age (in days, scaled) | −0.08 | 0.06 | −0.19 to 0.03 | 0.159 | 0.01 |
| Sex (reference level: female) | 0.06 | 0.12 | −0.17 to 0.29 | 0.608 | 0.04 |
| Face orienting (proportion of first look at the face) | 0.07 | 0.05 | −0.04 to 0.17 | 0.215 | 0.02 |
| Preference for face (proportion on face) | <0.01 | 0.06 | −0.11 to 0.11 | 0.935 | 0.05 |
| No. of objects explored (in 0–10 s) | 0.04 | 0.05 | −0.05 to 0.14 | 0.377 | 0.03 |
| **ECBQ self-regulation at 24 months** | **β** | **s.e.** | **95% CI** | **Uncorrected P value** | **FDR threshold** |
| Age (in days, scaled) | 0.05 | 0.07 | −0.09 to 0.19 | 0.475 | 0.03 |
| **Sex (reference level: female)** | **−0.38** | **0.13** | **−0.63 to −0.13** | **0.003** | **0.01** |
| Face orienting (proportion of first look at the face) | 0.06 | 0.06 | −0.04 to 0.17 | 0.248 | 0.02 |
| Face preference (proportion on face) | 0.04 | 0.06 | −0.08 to 0.15 | 0.548 | 0.05 |
| No. of objects explored (in 0–10 s) | 0.03 | 0.05 | −0.07 to 0.14 | 0.530 | 0.04 |
| **ITC social communication at 14 months** | **β** | **s.e.** | **95% CI** | **Uncorrected P value** | **FDR threshold** |
| Age (in days, scaled) | 0.07 | 0.07 | −0.07 to 0.21 | 0.348 | 0.04 |
| Sex (reference level: female) | −0.27 | 0.12 | −0.51 to −0.03 | 0.030 | 0.02 |
| Face orienting (proportion of first look at the face) | −0.01 | 0.06 | −0.12 to 0.11 | 0.918 | 0.05 |
| Face preference (proportion on face) | 0.13 | 0.06 | 0.02 to 0.24 | 0.026 | 0.01 |
| No. of objects explored (in 0–10 s) | 0.10 | 0.05 | 0.01 to 0.20 | 0.031 | 0.03 |
| **CDI receptive vocabulary at 14 months** | **β** | **s.e.** | **95% CI** | **Uncorrected P value** | **FDR threshold** |
| **Age (in days, scaled)** | **0.15** | **0.06** | **0.04 to 0.27** | **0.007** | **0.01** |
| **Sex (reference level: female)** | **−0.34** | **0.13** | **−0.59 to −0.09** | **0.008** | **0.02** |
| Face orienting (proportion of first look at the face) | −0.01 | 0.05 | −0.11 to 0.08 | 0.809 | 0.05 |
| **Face preference (proportion on face)** | **0.14** | **0.06** | **0.03 to 0.25** | **0.014** | **0.03** |
| No. of objects explored (in 0–10 s) | 0.07 | 0.05 | −0.02 to 0.16 | 0.142 | 0.04 |
| **CDI expressive vocabulary at 24 months** | **β** | **s.e.** | **95% CI** | **Uncorrected P value** | **FDR threshold** |
| **Age (in days, scaled)** | **0.21** | **0.05** | **0.11 to 0.31** | **<0.001** | **0.02** |
| **Sex (reference level: female)** | **−0.58** | **0.14** | **−0.85 to −0.32** | **<0.001** | **0.01** |
| Face orienting (proportion of first look at the face) | −0.01 | 0.05 | −0.11 to 0.09 | 0.872 | 0.05 |
| Face preference (proportion on face) | 0.04 | 0.06 | −0.08 to 0.15 | 0.550 | 0.04 |
| No. of objects explored (in 0–10 s) | 0.10 | 0.05 | <0.01 to 0.20 | 0.046 | 0.03 |

For each model, all predictors were entered together; hence statistics represent unique contributions for each predictor. Adjustments were made for multiple comparisons using the FDR step-up approach. s.e., standard error. In bold: significant predictors (P threshold in the FDR threshold column).

alternative related measures (that is, latency to look at the face; Supplementary Method 2). Given the adaptive and survival value of face orienting, it is not surprising that there is limited genetic variation linked to this phenotype. The small genetic variation associated with this phenotype might be related to face-selective processes as well as differences in general attention abilities[1].

For efficiency of visual exploration, indexed by the number of objects (including face and non-face ones) an infant looked at during the first 10 s of the trial, we did not find evidence for familial effects (genetic or shared environment), and variability was best explained solely by unique environmental factors (which include measurement error). Perhaps our participants were too young to be displaying a stable measure of exploration, as indeed exploratory gaze patterns have been shown to be less consistent in infancy[27], and/or our study was underpowered to detect subtle familial influences in this case (for details on power analysis in the study, see Supplementary Method 6).

Face orienting and face preference were moderately correlated, and there was evidence for shared genetic influences on face preference from face orienting (Fig. 2). While initial orienting and sustained attention to faces are hypothesized to be dissociated in terms

of underlying brain networks (subcortical versus cortical networks[15]), it is possible that the observed shared variance is driven by subcortical processes influencing both phenotypes at this age[28], by early emerging face-specific cortical structures influencing both face preference and orienting[29], or by an inflation of the co-variance due to potential dependency of the measures (where you look first will probably influence to some extent your preferences at longer timescales).

Relatedly, we found that face preference was phenotypically and aetiologically largely independent from another heritable social looking phenotype in infancy: eye-versus-mouth looking[9]. This dissociation shows that social looking is not a unitary phenomenon, but is composed of multiple phenotypically and aetiologically distinct subdimensions[1,10]. Additionally, sensitivity analyses focusing on orienting to and preference for the second most looked at object (car) suggested no genetic effects for 'car looking' (Supplementary Method 4), supporting the idea that the genetic effects observed in the main analyses may be specific to the social/non-social contrast.

We predicted that both face preference and efficiency of visual exploration would be associated to later development (language/social communication and self-regulation, respectively). However, only

the association between face preference and receptive vocabulary at 14 months was significant when applying stringent statistical criteria (Table 5, although it was not significant when controlled for gestational age instead of chronological age; Supplementary Method 8). We did not find an association between face preference and expressive vocabulary at 24 months. This could reflect the differences between the two scales (for further information, see Methods), but also equifinal developmental pathways[30] where a temporary disadvantage in language development in some children (those looking less at faces at 5 months) disappears over time.

We did not find any associations between looking measures and genome-wide polygenic scores for autism or ADHD. While this might be reflecting true null effects, it is possible that the current polygenic scores do not yet have enough predictive power to detect these links.

The study has some notable limitations. First, the number of face pop-out trials in our study was six and increasing the number of trials could potentially lead to more stable measures of infant face orienting and objects exploration (though probably at a cost of increased participant attrition). Second, while the BATSS study included almost 30% of the same-sex twin population in the area, it reflected families with a higher socioeconomic status (SES) compared with the Stockholm normative population[23]. This needs to be considered in the generalizability of our results as genetic and environmental estimates may vary in samples where SES has a wider distribution[31]. Third, in contrast to the objective assessment of gaze behaviours at 5 months, later development was only assessed via questionnaires to parents. Finally, the current study used static images, and generalizability to dynamic or real-life stimuli is not known[1,32].

In conclusion, our findings inform us about the aetiological influences on several important looking behaviours emerging early in infancy, and their developmental associations in the first 2 years of life. The results suggest two forms of gene–environment interplay unfolding at a micro-level in infancy. Firstly, because selective attention influences the input received, heritable preferences in infancy can be seen as a selection of the environment rooted in the individual's biology. Further, looking at faces may cause reactions from the social partner. Both will have cascading effects in cognitive and social development.

## Methods

Three hundred and eleven families of same-sex twins were recruited to the BATSS[23] and participated in an initial in-person assessment at 5 months old at Karolinska Institutet (data collection from April 2016 to February 2020) and participated in multiple follow-up online questionnaires at 14 months and 24 months (and 36 months, ongoing data collection at the time of submission of this work). BATSS was approved by the Regional Ethical Review Board in Stockholm and was conducted in accordance with all relevant ethical regulations and the Declaration of Helsinki. Parents gave informed consent to take part at each time point. A gift voucher of approximately €80 was given to each family at the initial in-person assessment. The main project sample description and inclusion criteria are described elsewhere[23]. Zygosity was estimated on the basis of DNA sampled from all infants. The current report uses a similar sample and the same twin statistical tools as a previous sample from our group[33].

We excluded 28 twins from the total BATSS sample due to twin-to-twin transfusion syndrome, seizures at the time of birth, very low birth weight (<1.5 kg) or spina bifida[23] (all parent reported). Furthermore, we excluded 23 infants because they did not complete the eye-tracking battery (technical reasons, time constraints, bad calibration or tiredness). Of the infants that completed the session, 35 infants did not provide enough valid trials and were excluded (valid trial criteria below). The final sample consisted of 536 infants (285 pairs with at least one individual with valid data; 251 pairs with valid data from both). The excluded (on the basis of invalid trials or not completing the eye-tracking assessment) and included infants did not statistically differ in terms of either parental education level, family income, sex or age.

### Eye-tracking protocol

To record infants' gaze a Tobii TX300 eye-tracker was used (sampling rate of 120 Hz) with MATLAB (version R2013b, MathWorks) and Psychtoolbox (for stimuli presentation, version 3.0.12) with custom algorithms written for the Eurosibs study[34]. The task battery started with an initial five-point calibration and was followed by rotations of free-viewing of the face pop-out task (see below), dynamic scenes trials (mixture of social and abstract content[35]), gaze-contingent gap-overlap trials[36], pupillary light reflex measurements[33], and post-calibration sequences; the task battery lasted for about 10 min.

**The face pop-out task.** The face pop-out task[5,6,11,34,37] was used to measure the various attentional processes involved in visual attention. It consisted of the presentation (fixed order, 20 s each) of a set of six different complex displays of objects, including a face (with direct eye gaze, three males and three females, counterbalancing ethnicity, and location of the face within the array) and four non-face competitors (including a 'noise' stimulus generated from the same face, a mobile phone, a bird and a car) (Supplementary Fig. 1). This type of display and the relatively long presentation time allow the study of the variation in active seeking of social information and attention flexibility by estimating the timing and preferential attention to faces and non-face stimuli.

Before each stimulus display, a small animation was presented and gaze-contingent methods started the presentation of the displays synchronous to the infant's look at the central animation, ensuring the gaze was at the centre of the screen at the start of the trial. One infant viewed seven trials of the face pop-out (the first trial was repeated because the protocol had to be restarted)—all trials were included in the analysis.

### Computation of primary measures

Gaze data for each pop-out trial was processed using custom-written MATLAB scripts (analysis steps and areas of interest (AOIs), in Supplementary Method 1 and Supplementary Fig. 2). Any trials with a proportion of valid (non-missing) data less than 0.25 (25%), with the total duration of data for a trial less than 5 s, or where no look at an AOI was made, were excluded. Measures were averaged across trials for each infant if at least four valid trials were found. The distributions and boxplots for the proportion of valid trials where each AOI was the first AOI looked at and the proportion of looking time to each AOI (relative to all AOIs) can be seen in Supplementary Fig. 3.

Face orienting was operationalized as the proportion of first looks at the face (that is, the number of valid trials where the face was the first object looked at in relation to the number of valid trials), in deviation of the pre-registered plan of using a composite measure of the proportion of first looks at faces and the mean latency to look at a face. This decision was based on a modest correlation found between latency and proportion of first look at the face (standardized $\beta = -0.33$, $P < 0.001$) and unmet twin assumptions for the composite measure (driven by unmet equality of means and variances across zygosity and twin order for latency). However, sensitivity analyses with this composite led to a similar pattern of genetic univariate and bivariate findings (Supplementary Method 2).

Face preference was operationalized as the mean ratio of looking at the face, that is, the sum of looking time at the face AOI divided by the sum of looking time at all AOIs averaged across valid trials.

Efficiency of object exploration was operationalized as the mean number of objects looked at, averaged across valid trials. Each array of objects was presented for 20 s, a longer duration than in previous studies' protocols[5,6,37] where shorter versions were used (12 and 15 s). The longer duration meant that it was likely that infants looked at the five objects during the trial (in 45% of all trials the five objects were looked at). For this reason, and in deviation of the pre-registered plan,

object exploration was estimated on the basis of only the first 10 s of the trial (in 22% of all trials, the five objects were looked at). The cut-off did not influence the results (Supplementary Method 3).

An analysis to contrast face orienting and preference (which reflect social versus non-social preferences) to the most attended (salient) non-social object (car) in the pop-out task, is reported in Supplementary Method 4.

**Gaze quality measures.** To control for potential effects of gaze quality in analyses, we estimated two gaze quality variables: the average proportion of missing data in the task (operationalized as the ratio of missing gaze per total data collected, averaged across valid trials) and the number of valid trials.

### Genome-wide polygenic scores

Genotyping of DNA samples was done using Infinium Global Screening Array (Illumina). Processing and quality control were done based on standard procedures and are described elsewhere[23]—for more details, see Supplementary Method 5. Polygenic scores were computed using the polygenic prediction via Bayesian regression and continuous shrinkage priors method[38], based on the most recent and largest (at the time of calculation of the scores, November 2020–March 2021) genome-wide association studies for ADHD[39], autism[40], bipolar disorder[41], major depressive disorder[42] and schizophrenia[43]. For this analysis, the first ten principal components of ancestry were included as covariates.

### Parent-rated developmental questionnaires

**Social communication.** The Communication and Symbolic Behavior Scales Developmental Profile Infant Toddler Checklist (ITC[44]) was used to measure socio-communicative behaviours (as indexed by the total raw score) at 14 months. A lower score is indicative of communication difficulties.

**Language.** The Swedish Early Communicative Development Inventory (CDI[45,46]), adapted from the Macarthur-Bates Communicative Development Inventory, was used to measure vocabulary at 14 months (the words and gestures form) and 24 months (the words and sentences form). In line with our study of eye-versus-mouth looking with the same sample[9], we used receptive vocabulary (number of words the child understands) at 14 months, and we used expressive vocabulary (number of words the child understands and says) at 24 months. At 14 months, the production scale produces substantial floor effects. At 24 months, the CDI reliably measures individual differences in language production and infants' receptive vocabulary is typically too large to be quantified by parents, and is not included in the words and sentences form.

**Self-regulation.** The Early Childhood Behavior Questionnaire (ECBQ[47]) was used to measure self-regulation (as indexed by the effortful control scale of the questionnaires) at 14 months (short-form, 107 items) and 24 months (very-short-form, 36 items).

### Analysis plans

An analysis plan for this study was registered in Open Science Framework[19] on 20 August 2021 (before data cleaning and analysis). R software (version 4.0.0) was used for all data computation and analyses. A power analysis was conducted before the data collection (Supplementary Method 6). All statistical testing were two-sided.

To test face orienting and preference against chance level, one-sample two-tailed tests were conducted for twins independently (Wilcoxon signed rank test was used for face orienting and $t$-test for face preference; because only the latter followed a normal distribution tested with the Shapiro–Wilk normality test). Effect sizes were estimated on the basis of mean minus the chance level (0.2) divided by the standard deviation (s.d.).

For twin models, the OpenMx package[48] (version 2.18.1 with NPSOL optimizer) with full-information maximum likelihood estimation was used, which allows for partially complete pairs (one twin missing) to be included.

For each looking measure, both saturated models (which test for the assumptions of equality of mean and variances across twin order and zygosity) and univariate twin models were fitted separately and reported in Supplementary Tables 1–6. A bivariate saturated model (which tested the assumption of equal phenotypic and CTCT correlations by zygosity and between twins) and a bivariate twin model (Cholesky decomposition) were fitted on the two variables, namely face orienting and face preference. As noted in the main text, twin structural equation model-fitting is a statistical approach involving the decomposition of variance in a phenotype/set of phenotypes into genetic (A), shared environment (C) and unique environment (E) influences. A full model (including A, C and E) was evaluated against several possible nested (simpler) models[49]. The best-fitting nested model was defined as the non-significant model with the lowest AIC value. When a nested model is significant, it means that it has poorer fit than the full model, indexed by the $\chi^2$ distribution, and hence should be excluded (this entails that the selected nested model is always statistically non-significant[50]). The AIC fit statistic incorporates information about both explained variance and parsimoniousness; the lowest value corresponds to the best model. Twin and CTCT correlations were derived from the constrained saturated models, in which means, variances, phenotypic and CTCT correlations were constrained to be equal across twin order and zygosity. When the pattern of correlations suggested non-additive genetic effects (D; MZ correlation more than twice the DZ correlation), a decision was made to report an ACE model (the model including A, C, and E) rather than an ADE model (the model including A, D, and E) to our data due to sample size (for the ADE bivariate model results, see Supplementary Table 8). In accordance with the standard reporting for twin research[51,52], we report CIs for each component included in the best-fitting model, whether the CI overlaps with zero shows whether the component is statistically significant.

Association analyses were conducted, whenever possible, with the whole twin sample (that is, including both twins in a pair, including pairs with one twin missing) using linear regression models implemented as generalized estimating equations (GEEs; using the drgee package[53]), with cluster-robust standard errors to account for family relatedness, to derive $\beta$ estimates and $P$ values. All measures were scaled so that $\beta$ estimates were standardized. Effect sizes ($\Delta R^2$) were calculated on the basis of comparing the $R^2$ of the null model (that is, the model with only covariates included) and of corresponding models. When controlling for multiple testing (when testing the longitudinal phenotypic associations) a false discovery rate (FDR) step-up approach was used across analyses using the same outcome.

Chronological age (in days) and sex were always included in twin models and added to the GEE models as covariates. Associations between the gaze quality covariates (proportion missing gaze and number of valid trials) and the gaze-based primary visual attention measures were tested within the GEE framework (one linear model with both covariates as predictors were run for each primary variable). If statistically significant, the gaze quality measures were regressed out from the main dependent variables before all other analyses. Eye-tracking accuracy and precision were also tested as additional gaze quality covariates in a sensitivity analysis presented in Supplementary Method 7. Corrected age (age estimated on the basis of birth date and gestational age at birth) was included in twin models and added to the GEE models as covariates (in replacement of chronological age) in a sensitivity analysis reported in Supplementary Method 8.

### Reporting summary

Further information on research design is available in the Nature Portfolio Reporting Summary linked to this article.

## Data availability

Unrestricted sharing of pseudonymized personal data was not specified in the study ethics application; hence, data are not uploaded to a public repository. However, data are available from T.F.Y. (terje.falck-ytter@psyk.uu.se) on reasonable request. Request will be responded to within 1 week. Sharing pseudonymized (coded) data from the study will require a data sharing agreement according to Swedish and EU law.

## Code availability

The face pop-out pre-processing workflow (implemented in MATLAB and described in Supplementary Method 1) is part of a shared agreement and available from L.M. on reasonable request. The statistical analysis scripts (implemented in R) are publicly available in OSF (https://osf.io/zseh2/). The R code for the raincloud plot visualizations (Fig. 1 and Supplementary Fig. 3) has been adapted from Allen et al.[54].

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

## Acknowledgements

We thank all participating families, as well as the BATSS testing team: L. Hamrefors, M. Siqueiros Sanchez, J. Hättestrand, L. Myers, J. Kronqvist, S. Jönsson, A. Kernell, C. Schreiner, S. Lingö, A. Liljebäck, I. Enedahl, M. Andreasson, L. Belfrage, M. Savallampi, I. Ocklind, H.N. Norrman and I. Shragge. Special thanks to D. Li for contributing to the genetic data processing, and to M. Rudling for contributing to the CDI data processing and the discussion around the associations with language developmental outcomes. We also thank all the Development and Neurodiversity Lab members (especially G. Bussu, L. Andersson Konke and I. Hardiansyah), and A.H. Henriksen, for valuable discussions and feedback. The images/stimuli used in the face pop-out experimental task (and shown in Supplementary Figs. 1 and 2) were created at Birkbeck, University of London. We have been given permission to use and publish them. This research was supported by the Stiftelsen Riksbankens Jubileumsfond (NHS14-1802:1 to T.F.Y. and Pro Futura Scientia in collaboration with SCAS to T.F.Y.), the Swedish Research Council (2018-06232 to T.F.Y.), the Knut and Alice Wallenberg foundation (to T.F.Y.), and the European Union (EU-MSCA Initial Training Networks: 642996, BRAINVIEW to T.F.Y. and 814302, SAPIENS to T.F.Y.). K.T. was supported by the Swedish Foundation for Strategic Research (FFL18-0104) and the Swedish Brain Foundation (FO2021-0073). L.M. was supported by the Innovative Medicines Initiative Joint Undertaking (under grant agreement no. 115300, resources of which are composed of financial contribution from the European Union's Seventh Framework Programme (FP7/2007—2013) and EFPIA companies' in kind contribution), the Innovative Medicines Initiative 2 Joint Undertaking (under grant agreement no. 777394, which receives support from the European Union's Horizon 2020 research and innovation programme), EFPIA and AUTISM SPEAKS, Autistica, SFARI, the Medical Research Council (MR/K021389/1 and MR/T003057/1), and the NIHR Maudsley Biomedical Research Centre at South London and Maudsley NHS Foundation Trust and King's College London. We acknowledge the KI Biobank for handling the biological samples, SNP&SEQ Technology Platform, Uppsala University, for genotyping, and the Swedish National Infrastructure for Computing (SNIC) at UPPMAX, partially funded by the Swedish Research Council through grant agreement no. 2018-05973, for computations. Any views expressed are ours and not necessarily those of the funders. The funders had no role in study design, data collection and analysis, decision to publish or preparation of the paper.

## Author contributions

Conception and design: A.M.P., T.F.Y., A.R. and K.T.; data preparation: A.M.P., L.M. and K.T.; analysis of data: A.M.P. with contributions of T.F.Y. and M.J.T.; writing (original draft): A.M.P. and T.F.Y., with contributions from M.J.T. and C.V.; and writing (critical review and editing): all authors.

## Funding

## Competing interests

The authors declare no competing interests.

## Additional information

**Correspondence and requests for materials** should be addressed to Ana Maria Portugal or Terje Falck-Ytter.

# Reporting Summary

## Statistics

For all statistical analyses, confirm that the following items are present in the figure legend, table legend, main text, or Methods section.

| n/a | Confirmed | |
|---|---|---|
| ☐ | ☒ | The exact sample size (*n*) for each experimental group/condition, given as a discrete number and unit of measurement |
| ☐ | ☒ | A statement on whether measurements were taken from distinct samples or whether the same sample was measured repeatedly |
| ☐ | ☒ | The statistical test(s) used AND whether they are one- or two-sided *Only common tests should be described solely by name; describe more complex techniques in the Methods section.* |
| ☐ | ☒ | A description of all covariates tested |
| ☐ | ☒ | A description of any assumptions or corrections, such as tests of normality and adjustment for multiple comparisons |
| ☐ | ☒ | A full description of the statistical parameters including central tendency (e.g. means) or other basic estimates (e.g. regression coefficient) AND variation (e.g. standard deviation) or associated estimates of uncertainty (e.g. confidence intervals) |
| ☐ | ☒ | For null hypothesis testing, the test statistic (e.g. *F*, *t*, *r*) with confidence intervals, effect sizes, degrees of freedom and *P* value noted *Give P values as exact values whenever suitable.* |
| ☒ | ☐ | For Bayesian analysis, information on the choice of priors and Markov chain Monte Carlo settings |
| ☒ | ☐ | For hierarchical and complex designs, identification of the appropriate level for tests and full reporting of outcomes |
| ☐ | ☒ | Estimates of effect sizes (e.g. Cohen's *d*, Pearson's *r*), indicating how they were calculated |

*Our web collection on statistics for biologists contains articles on many of the points above.*

## Software and code

Policy information about availability of computer code

| | |
|---|---|
| Data collection | Gaze recording and experiment presentation was done using MATLAB (version R2013b, MathWorks, Natick, MA, USA), Psychtoolbox (version 3.0.12), and custom algorithms (part of a shared agreement and available from co-author L.M. on a reasonable request). |
| Data analysis | R software (version 4.0.0) was used for all data computation and analyses (the OpenMx package (version 2.18.1) was used for twin analyses and the drgee package for GEE analyses). The scripts are publicly available in OSF (https://osf.io/zseh2/?view_only=1d7a815ff87148a6af5a6b58c427419c). |

For manuscripts utilizing custom algorithms or software that are central to the research but not yet described in published literature, software must be made available to editors and reviewers. We strongly encourage code deposition in a community repository (e.g. GitHub). See the Nature Portfolio guidelines for submitting code & software for further information.

## Data

Policy information about availability of data

All manuscripts must include a data availability statement. This statement should provide the following information, where applicable:
- Accession codes, unique identifiers, or web links for publicly available datasets
- A description of any restrictions on data availability
- For clinical datasets or third party data, please ensure that the statement adheres to our policy

Unrestricted sharing of pseudonymized personal data was not specified in the study ethics application, hence data are not uploaded to a public repository. However, data are available from Terje Falck-Ytter (terje.falck-ytter@psyk.uu.se) on a reasonable request. Request will be responded to within 1 week. Sharing pseudonymized (coded) data from the study will require a data sharing agreement according to Swedish and EU law.

# Field-specific reporting

Please select the one below that is the best fit for your research. If you are not sure, read the appropriate sections before making your selection.

☐ Life sciences    ☒ Behavioural & social sciences    ☐ Ecological, evolutionary & environmental sciences

For a reference copy of the document with all sections, see nature.com/documents/nr-reporting-summary-flat.pdf

# Behavioural & social sciences study design

All studies must disclose on these points even when the disclosure is negative.

| | |
|---|---|
| Study description | The study involved quantitative research methodologies. It used a classic twin design (i.e., compare similarity in monozygotic and dizygotic twin pairs) and structural equation model fitting approach to study individual differences in looking behaviours to faces vs non-face objects in infancy. |
| Research sample | The sample included 536 5-month-old same-sex twins (251 females, 293 monozygotic, mean age = 168 days), recruited from the greater Stockholm area in Sweden for the Babytwins Study Sweden (see Falck-Ytter et al, The Babytwins Study Sweden (BATSS): A Multi-Method Infant Twin Study of Genetic and Environmental Factors Influencing Infant Brain and Behavioral Development. Twin Res Hum Genet, 2021. 24(4): p. 217-227). |
| Sampling strategy | Same sex twin families living in the Stockholm area were identified via the Swedish Population Registry, and invited to participate via letters and telephone calls. In total, 1068 families were invited to join the study, of which 311 families participated in the study (n = 622 infants). The pre-established target sample size was 620 individuals (310 pairs) based on the size of previous twin studies with toddlers (e.g., Ronald et al., Exploring the relationship between autistic-like traits and ADHD behaviors in early childhood: Findings from a community twin study of 2-year-olds. Journal Of Abnormal Child Psychology, 2010. 38, 185–196) and informed by a general power calculation (see Supplementary Information). For more information about the study see Falck-Ytter et al, The Babytwins Study Sweden (BATSS): A Multi-Method Infant Twin Study of Genetic and Environmental Factors Influencing Infant Brain and Behavioral Development. Twin Res Hum Genet, 2021. 24(4): p. 217-227. |
| Data collection | Parents gave informed consent to take part at each time point. A gift voucher of approximately 80€ was given to each family in the first lab-assessment. Data was collected using an eye-tracking device at 5 months of age, and parent-rated on-line questionnaires at 5 months, 14 months, and 24 months. Saliva samples were also collected from infants by research assistants. The research assistants collecting the data were blind to the zygozity of the twin pairs as well as the experimental hypotheses. |
| Timing | The first lab-assessment (at 5 months) was collected from April 2016 to February 2020. |
| Data exclusions | Participants were excluded due to pre-established exclusion criteria (seizures at the time of birth, spina bifida, twin-to-twin transfusion syndrome, birthweight below 1.5 kg; n = 28 infants). Further, some infants did not complete the eye-tracking assessment due to technical reasons, time constraints, bad calibration, or tiredness (n = 23 infants), and some did not have enough valid data in the experimental task (n = 35 infants). |
| Non-participation | From the target population (see Recruitment section below) 29% of families ultimately participated in the lab-assessment at 5 months. At 14 months, 86% of participating families provided data for at least one questionnaire. At 24 months, 72% of families provided data for at least one questionnaire. |
| Randomization | Participants were not allocated into experimental groups. |

# Reporting for specific materials, systems and methods

We require information from authors about some types of materials, experimental systems and methods used in many studies. Here, indicate whether each material, system or method listed is relevant to your study. If you are not sure if a list item applies to your research, read the appropriate section before selecting a response.

## Materials & experimental systems

| n/a | Involved in the study |
|---|---|
| ☒ | Antibodies |
| ☒ | Eukaryotic cell lines |
| ☒ | Palaeontology and archaeology |
| ☒ | Animals and other organisms |
| ☐ | ☒ Human research participants |
| ☒ | Clinical data |
| ☒ | Dual use research of concern |

## Methods

| n/a | Involved in the study |
|---|---|
| ☒ | ChIP-seq |
| ☒ | Flow cytometry |
| ☒ | MRI-based neuroimaging |

# Human research participants

Policy information about studies involving human research participants

| | |
|---|---|
| Population characteristics | See above. |
| Recruitment | Same sex twin families living in the Stockholm area were identified via the Swedish Population Registry, and invited to participate via letters and telephone calls. 29% of the target population ultimately participated in the BATSS study.  Possible self-selection sources include socio-economic status, ethnic background, and physical or mental health issues in the parents. If and how such factors affect looking preferences in infancy is not known. For more information about the sample and possible biases, see Falck-Ytter et al, The Babytwins Study Sweden (BATSS): A Multi-Method Infant Twin Study of Genetic and Environmental Factors Influencing Infant Brain and Behavioral Development. Twin Res Hum Genet, 2021. 24(4): p. 217-227. |
| Ethics oversight | The study was approved by the Regional Ethical Review Board in Stockholm. |

Note that full information on the approval of the study protocol must also be provided in the manuscript.

