## [Peer Review File · Nature Human Behaviour]

Peer Review Information

Journal: Nature Human Behaviour

Manuscript Title: Infants' looking preferences for social versus non-social objects reflect genetic variation

Corresponding author name(s): Ana Portugal

Reviewer Comments & Decisions:

Decision Letter, initial version:
--

1st June 2022

Dear Dr Portugal,

Thank you once again for your manuscript, entitled "Infants' looking preferences for social versus non-social objects reflect genetic variation and are linked to later language development," and for your patience during the peer review process.

Your manuscript has now been evaluated by 3 reviewers, whose comments are included at the end of this letter. Although the reviewers find your work to be of interest, they also raise some important concerns. We are interested in the possibility of publishing your study in Nature Human Behaviour, but would like to consider your response to these concerns in the form of a revised manuscript before we make a decision on publication.

To guide the scope of the revisions, the editors discuss the referee reports in detail within the team, including with the chief editor, with a view to (1) identifying key priorities that should be addressed in revision and (2) overruling referee requests that are deemed beyond the scope of the current study. We hope that you will find the prioritised set of referee points to be useful when revising your study. Please do not hesitate to get in touch if you would like to discuss these issues further.

1) Reviewer 1 highlights a key issue with your analyses which undermine the conclusions that can be drawn about social (faces) vs non-social objects. Please follow the reviewer's recommendation to carry out symmetric analyses examining first look and sustained attention for both faces and objects.

2) The reviewers raise a number of important points regarding your statistical analyses and interpretation, including the lack of statistically supported comparisons in some cases and the inappropriate interpretation of results that did not meet the threshold for declaring significance. Please note that we do not allow the interpretation of statistically non-significant results, no matter how close

to the significance threshold they may be. From an editorial perspective, we also ask that all null results that are theoretically interpreted are done so after applying appropriate statistical tools to quantify support for the null hypothesis, such as Bayes Factors (for guidance, please see <https://www.frontiersin.org/articles/10.3389/fpsyg.2014.00781/full>).

3) Reviewer 3 points out that novelty claims are inappropriate given prior work, especially your own. Please also note that as a matter of policy, we ask that authors refrain from making novelty or priority claims (except in cases of discovery of new loci in GWAS studies of a specific phenotype). In revision, we ask that you remove novelty claims and contextualize your study appropriately within the existing body of published work, without overstating their significance.

In sum, we invite you to revise your manuscript taking into account all reviewer and editor comments. We are committed to providing a fair and constructive peer-review process. Do not hesitate to contact us if there are specific requests from the reviewers that you believe are technically impossible or unlikely to yield a meaningful outcome.

We hope to receive your revised manuscript within three months. I would be grateful if you could contact us as soon as possible if you foresee difficulties with meeting this target resubmission date.

- Include a "Response to the editors and reviewers" document detailing, point-by-point, how you addressed each editor and referee comment. If no action was taken to address a point, you must provide a compelling argument. When formatting this document, please respond to each reviewer comment individually, including the full text of the reviewer comment verbatim followed by your response to the individual point. This response will be used by the editors to evaluate your revision and sent back to the reviewers along with the revised manuscript.
- Highlight all changes made to your manuscript or provide us with a version that tracks changes.

[REDACTED]

We look forward to seeing the revised manuscript and thank you for the opportunity to review your work. Please do not hesitate to contact me if you have any questions or would like to discuss these revisions further.

Sincerely,

Charlotte Payne

Charlotte Payne, PhD
Senior Editor
Nature Human Behaviour

Reviewer expertise:

Reviewer #1: Developmental neuroscience; social cognition; attention in infants; autism

Reviewer #2: Developmental neuroscience; autism; infant attention

Reviewer #3: Autism genetics

REVIEWER COMMENTS:

Reviewer #1:

Remarks to the Author:

Key results

This study examined attention to onscreen social (via initial orienting to faces and sustained face preference) as well as object (via # of objects explored) features in a sample of 5-month-old twins. They found that both characteristics of social attention were heritable (although face preference more than face orienting) and that shared environmental contributions were non-significant. Sustained face preference was associated with later receptive (but not expressive) vocabulary but neither measure of social attention was associated with polygenic risk scores for ASD or ADHD or future self-regulation or social communication. There were no twin-level concordances between exploration of stimulus items.

Validity

One aspect of the study design raises questions about validity. Given the study's title and goal to "experimentally dissociate looking behavior to social (faces) versus non-social objects" and quantifying genetic contributions to both, it is unclear as to why the authors did not conduct symmetric analyses examining first look and sustained attention to faces as well as to objects. This approach, followed by measures of the heritability for each, would strengthen the design of this study, better validate the cross-phenotype correlations, and potentially provide stronger evidence for genetic contributions to social but not non-social viewing. What was the rationale for instead simply quantifying numbers of stimuli explored?

Originality & significance

I find the goals of this study to be important for our understanding of social cognitive development

and I think the results from this unique sample are of immediate interest. I find the results about the heritability of social attention processes, and their relationship with future receptive language capacities, to be compelling and scientifically interesting, although it is at present unclear to me if these same effects exist for the processing of objects (given the non-comparable analyses).

Data & methodology

- The "CTCT" acronym should be defined in the main body of the text
- Please elaborate on what "twin assumptions" refer to on p. 10 line 121?
- In addition to the reported gaze quality measures (missing data and number of valid trials), and in accordance with standard practice, it would be helpful to know eye-tracker calibration accuracy and precision for the sample. Did these metrics influence any of the measures?
- Please use age corrected for gestational duration as there is often increased variability in gestational duration among twins.

Preregistration

This was a pre-registered project and the authors make note of deviations from the pre-registration and report registered analyses in the Supplement.

Appropriate use of statistics and treatment of uncertainties

(As noted above):

Given the goal of "experimentally dissociate looking behavior to social (faces) versus non-social objects" and quantifying genetic contributions to both, I am unclear as to why the authors did not conduct symmetric analyses examining first look and sustained attention to faces as well as objects. This approach, followed by measures of the heritability for each, would strengthen the design of this study, better validate the cross-phenotype correlations, and potentially provide stronger evidence for genetic contributions to social but not non-social viewing. What was the rationale for instead simply quantifying numbers of objects attended to?

Custom code

R code is available on OSF.

Conclusions

- I find it difficult to interpret results from the non-social object findings given that the analyses were not comparable to those of the faces
- Should "face" be included in the sentence on line 206 p. 16?
- The logic of the sentence on p. 17 lines 230-232 is unclear to me. What were the authors' initial hypotheses (and rationale) regarding this association?
- The Discussion would be strengthened by contextualizing the findings in relation to the authors' initial hypotheses.
- The authors should discuss how they interpret the fact that face preference was associated with later receptive but not expressive language.
- The authors should acknowledge the limitations of parent-report outcomes

Suggested improvements

- Rephrase lines 185-187 for clarity

Clarity & context

- The abstract should better reflect full extent of the findings from this study, inclusive of null findings. For instance, from the abstract, it is unclear what non-social objects were studied or the associated finding –these should be included as they were core aims of the study. Additionally, the null associations with polygenic risk scores should also be mentioned as well as the lack of associations with self-regulation or social communication.
- It is unclear what the authors' hypotheses were for the study –the manuscript would be strengthened by presentation of the authors' hypotheses.

Reviewer #2:

Remarks to the Author:

Key results

Thank you for the opportunity to read and review the manuscript titled "Infants' looking preferences for social versus non-social objects reflect genetic variation and are linked to later language development". This study aims to assess the extent to which individual differences in social attention to faces reflect genetic and environmental factors. The study also aims to understand associations between attention to faces and later developmental skills such as social communication, language, and self-regulation.

Overall, results indicated that orienting to faces as well as sustained attention to faces are heritable. These findings are unique and intriguing in that they consider the impact of genetics on an infant's visual environment. The findings from this manuscript also indicate that visual preference for faces is positively associated with downstream receptive vocabulary.

Validity

- It is perhaps important to discuss the ecological validity of the stimuli used. This research used a static array of images that have limited ecological validity as a social attentional measure. For instance, the gaze to faces can be impacted by the social content of face (see review by Hessels, 2020).

Significance

- This research is unique in that it considers the impact of genetic and environmental influences on attention to faces in a large sample of infants.

References

- The reference list is comprehensive and adequate for the topic.

Your expertise

- The reviewer are experts in language, social communication, and autism spectrum disorder. I do not have expertise in genetics and cannot comment on: the use of polygenic scores for autism/ ADHD, nor the justification for using proportion first looks to the face vs latency based on the unmet twin assumption

Providing constructive feedback

Data and methodology

- In the methods section, under gaze quality measures, did the authors collect accuracy data? This might be important to considering that eye tracking accuracy is typically poor in infants, and this might impact AOI-based detection of gaze location (Dalrymple et al., 2018).
- In the methods section, line 369, the authors state that they used the total score from the CSBS ITC. Was the standard score used here?
- The rationale to investigate CDI receptive vocab at 14 months and CDI expressive vocab at 24 months was not developed.
- Can the authors confirm that Table 1 presents data in the form of % and not proportions?
- The AOIs are quite large. Did the authors confirm that there is sufficient distance in between the AOIs? A common metric is to ensure that 1 degree visual angle.

Analytical approach

- Results section, Table 3: In this table, the 95% CI for the correlations for MZ and DZ twins overlap. However, the authors conclude that correlations were lower for DZ twins. There seems to be considerable variation in the correlations, which may impact drawing the conclusion that the groups correlations were significantly different. The authors should directly test the differences in a single model if they want to make such conclusions.
- Many twins are born prematurely. Do the authors use a chronological age corrected for gestational age?
- The type of FDR correction that used should be specified in the main manuscript, along with how the correction was applied (across all analyses, across analyses using the same DV, ect)
- Results from null hypothesis testing are either significant or they are not significant. Referencing results as "marginal" (line 138) is not appropriate.
- I commend the authors for reporting null results and for assessing group differences between infants that were excluded versus those that were included.

Clarity and context

- In the introduction, the authors note that attending to social vs nonsocial is not "straightforward". Can the author elaborate on why they think the process is not straightforward?
- The authors claim to look at ASD and ADHD "outcomes". This seems like an odd phrase given that they only looked at social communication (their autism "outcome") at 14 months of age. ASD is not often diagnosed until 2 years of age or much older, so claiming to measure an outcome at 14 months seems inappropriate. The authors should also be mindful of falling into trap that thinking measuring autism traits in the general population is a way to study autism (see Sasson and Bottema-Beutel, 2021, <https://doi.org/10.1177/13623613211058515>).
- In the discussion section, (line 241) the authors refer to a report of atypical engagement with social scenes in infants later diagnosed autistic. There has been considerable variability in the social attention literature for autistic infants (see Guillion et al., 2014 for a review on this topic). The authors

may consider including this in the discussion. I would caution against a statement based on just one report on this topic given the variability of findings.

- I wonder if the authors could edit the same sentence to improve clarity (discussion, line 240). The sentence starts off discussing the association between face preference and later social communication and language and report that this is in line with reports in autistic infants. Correct me if I am wrong, but the reference here, Chawarska (2013) did not measure associations to later language. Young et al. (2009) might be a useful reference here. Additionally, the authors state "which are in line with reports.." but provide a singular reference.

Suggested (minor) improvements

- Throughout the manuscript, where the author refers to statistical values, they need to be italicized. For instance "p-values", "R2"etc.

- Results section, paragraph 1: Can the authors include effect sizes for the t values? Similar comment for Table 5, results section.

Reviewer #3:

Remarks to the Author:

Review Portugal et al, Nat Hum Behav 19209_0

I appreciate the chance to read and review this paper analyzing genetic and other influences on infants' looking preferences. The authors have a very large study group of same-sex twins, and have both genomic data and eye-tracking assay measures. As one would expect, they find that infant social visual preferences have measurable heritability. These authors have used many of the heritability techniques in this paper previously, most notably in a 2021 JCPP paper looking at the same study sample but different scientific question. It is not surprising to me that there would be insignificant correlations between face orienting/preference and any of the polygenic scores calculated by the authors, even that for ASD, given the state of that research area.

Strengths:

Study has eye tracking data from one timepoint and parent-report questionnaire from three timepoints in the same cohort and have DZ and MZ twin pairs, along with genomic data.

Authors have analyzed influence of genetics vs environment on face orienting and face preference. This is a fascinating question.

Size of study group is commendably larger than previous studies in this arena.

Problems:

Context statements are not supported by the data. Lines 35-38: this is not the first study to show that individual perceptual choices are heritable. At a minimum, this must be reworded to reflect the novelty actually reported in the paper.

Similarly, lines 50-54 are just not correct. Reference #9, cited in this very paper, exactly looked at "to what extent visual preference for faces versus non-social information in infancy reflects genetic

variation in the population” as claimed in lines 51-52 here (see extended data figure 6, <https://pubmed.ncbi.nlm.nih.gov/28700580/#&gid=article-figures&pid=extended-data-figure-6-uid-5>) . In fact, that reference had MZ twins, DZ twins, and other pairs, which is a more thorough design than this study for asking the question about G vs E.

Insufficient details in the references, lines 356-364, on calculating polygenic risk scores. Those calculations involve multiple filters and decision points, and what’s reported now is not sufficient to aid replication or understanding in context. I see this is a quite similar methods paragraph to a previous paper by the authors, <https://acamh.onlinelibrary.wiley.com/doi/10.1111/jcpp.13564>, but with even fewer details.

(A more minor point: unfortunately, for most conditions, 2019 papers are not now the most recent/up to date GWAS. I understand the authors mean at the time when the calculations were being done, and they may want to more explicitly state that.)

I do not understand the choice made by the authors in table 3, to say that there are genetic influences on face orienting with values of 0.20 and 0.05 but NOT on object orienting, with values of 0.05 and 0.15. The range on the confidence intervals obviously overlaps there. I am definitely not a statistician so others may understand this choice, but I would suggest some more explanation at least, as the decision is then to not show any more analyses of object orienting.

Minor corrections/points:

Line 139 is not a complete sentence: “whereas the shared genetic influences were – Table 4.”

More description of the models/tests used in lines 119-122 would be helpful for non-specialists.

Line 211: please forgive me if I missed it, but I did not see power calculations included. Is this something the authors could provide, to support statements of being underpowered? I would consider this optional but nice to have.

The 2021 Portugal et al paper (congratulations to the authors!) in JCPP used such similar methods that I am a little surprised not to see it cited here.

Questions more out of curiosity:

Given the time which has elapsed since the study was conducted, it is likely that some of the participants would have received ASD/ADHD diagnoses at this point. For a reported prevalence of ~1% in Sweden overall and 3.66% in an immigrant group in Sweden (Linsand et al JADD 2021), the authors should see several twin pairs at least who have already been diagnosed. In fact this could have been reported at the 36-month questionnaire, of course. Can they quickly go into the study records and find out if this is the case? It would be fascinating to how genomic or eye tracking markers corresponded to later diagnosis of twin pairs.

I see that in reference 18, the ethnicity of this sample is described as “Visual inspection of the results of the PCA confirmed that the sample was largely homogenous in terms of Swedish/European ancestry, with only a minority of mixed genetic ancestry or non-European genetic ancestry.” To what extent would the authors expect to see these influences in their GxE calculations, given that E might be more noticeable in ASD studies than in those for other neurodevelopmental conditions? Have they accounted for this / can it be accounted for?

Author Rebuttal to Initial comments**Reviewer #1:**

Remarks to the Author:

Key results

This study examined attention to onscreen social (via initial orienting to faces and sustained face preference) as well as object (via # of objects explored) features in a sample of 5-month-old twins. They found that both characteristics of social attention were heritable (although face preference more than face orienting) and that shared environmental contributions were non-significant. Sustained face preference was associated with later receptive (but not expressive) vocabulary but neither measure of social attention was associated with polygenic risk scores for ASD or ADHD or future self-regulation or social communication. There were no twin-level concordances between exploration of stimulus items.

Validity

One aspect of the study design raises questions about validity. Given the study's title and goal to "experimentally dissociate looking behavior to social (faces) versus non-social objects" and quantifying genetic contributions to both, it is unclear as to why the authors did not conduct symmetric analyses examining first look and sustained attention to faces as well as to objects. This approach, followed by measures of the heritability for each, would strengthen the design of this study, better validate the cross-phenotype correlations, and potentially provide stronger evidence for genetic contributions to social but not non-social viewing. What was the rationale for instead simply quantifying numbers of stimuli explored?

RESPONSE #1. Thank you for this comment. Indeed, in principle, it is reasonable to suggest symmetrical analyses to 'complete the picture'. However, we want to start by clarifying that the stimuli used in our study (arrays with 5 objects) always contained both one social (the face) and four non-social objects, hence infants' looking duration to one aspect of the scene will necessarily affect their looking duration to other aspects. In other words, the primary variables (% trials and % of looking time) are really about preference for social VERSUS non-social objects, which is how 'social attention' is typically operationalized in the literature.

Thus, intuitively appealing as it is to separate social and non-social attention, it is not feasible and not our goal either. Rather, and in line with previous work on social attention and this specific stimulus paradigm, we contrast attention (looking) to social VERSUS non-social elements (when these are competing for attention at the same moment in time).

Specifically, first look was defined as the % of looks to faces relative to all objects in the stimulus. Thus, this measure already includes non-social objects in the calculation. **Hence, a symmetrical measure of this variable would be the inverse of the original measure (“% first look to non-social objects”, and its heritability would be the same).** Similarly, the second variable (preference for faces) was defined as the % to faces relative to all objects. Thus, this measure too is about looking at social objects (faces) relative to non-social objects (the remaining non-facial objects), **and a symmetrical non-social measure would be the inverse of the original measure, with the same heritability.** Lastly, and importantly, the total number of objects looked at was not meant to be a symmetrical non-social measure, but simply a measure tapping into how efficiently the infants explored the objects (social and non-social) in the scene.

We have gone through all descriptions of the stimuli in the Introduction and Methods and tried to clarify these important aspects, to avoid any misunderstanding about the goal of the study, the stimuli and task design, and the scope of our analysis.

We want to stress that the operationalizations used are not new to this study, previous work using the ‘face pop out task’ have all used the same or similar measures (Gliga et al., 2009; Elsabbagh et al., 2013; Hendry et al., 2018; Gui et al., 2020).

While we do not think it is possible, given the above arguments, to calculate independent and fully symmetric measures of social and non-social attention, the Reviewer comment prompted us to think about ways to investigate first looks and looking duration to non-social **salient** objects. These results are now reported in Supplementary Methods 4. Specifically, we noted that after face (looked at 44% of time with a relatively large variability), the most looked at object was the car (19% of time, with a smaller variability, see distribution in Figure S 2). Therefore, we computed car orienting (first looks to the car, relative to all non-social objects, i.e., excluding the trials where face was looked at first) and looking preference for the car (relative to all non-social objects, i.e. excluding looking time to the face). We believe these measures can be seen as approximately analogous to the two first face looking variables, and at the same time are mathematically independent of the two main variables of interest (face capture and face preference, as reported previously).

Using this approach, we found no significant genetic effects in terms of either car orienting (E model was the best fitting model) or car preference (the AE model was the best fitting model but the genetic effect estimate was not significant, $A = .14$, CI: 0-.29). In terms of the bivariate model between car orienting and car preference ($rPh = .25$, CI: .17-.33), the E model was the best fitting model, with most E on car preference being unique to that variable (unique $E = .94$,

CI: 0.89-0.97), and just a small significant proportion being shared with car orienting (shared $E = .06$, CI: .03-.11). We conclude that the genetic effects seen in face preference and orienting do not generalise to another object (the second most attended one), strengthening the argument that this pattern is specific to faces. We have added a sentence about this in the main text (Discussion, lines 213-216), linking to the Supplementary Methods 4.

Originality & significance

I find the goals of this study to be important for our understanding of social cognitive development and I think the results from this unique sample are of immediate interest. I find the results about the heritability of social attention processes, and their relationship with future receptive language capacities, to be compelling and scientifically interesting, although it is at present unclear to me if these same effects exist for the processing of objects (given the non-comparable analyses).

RESPONSE #2. Thank you for this positive evaluation. As noted in the previous Response, the heritability for looking to faces is for looking time to faces compared to other objects, hence the measure is not an isolated ‘social’ measure, but more about how infants prioritised between social and non-social objects.

Data & methodology

- The “CTCT” acronym should be defined in the main body of the text

RESPONSE #3. This has been fixed now.

- Please elaborate on what “twin assumptions” refer to on p. 10 line 121?

RESPONSE #4. Twin assumptions refer to assumptions that should be met in the data for formal twin modelling. These are assumptions of equality of means and variances across twin order and zygosity, and the assumptions of the equality of phenotypic and CTCTs correlations across twin order and zygosity. We had presented this in the methods section “Analysis plans”, but added them now also in the results section (lines 113-116).

- In addition to the reported gaze quality measures (missing data and number of valid trials), and in accordance with standard practice, it would be helpful to know eye-tracker calibration accuracy and precision for the sample. Did these metrics influence any of the measures?

RESPONSE #5. Thank you for your suggestion of checking accuracy and precision in the sample. These metrics have now been computed and tested in a similar way as it was done with missing data and number of valid trials in the previous version. Only accuracy was related significantly to face preference (but not with the other measures). A sensitivity analysis was performed for face preference after accuracy has been regressed from the

measure and this is reported now in Supplementary Methods 7. The results and conclusions remained the same.

- Please use age corrected for gestational duration as there is often increased variability in gestational duration among twins.

RESPONSE #6. We agree that it is important to consider corrected age given our twin sample (although please note that because we excluded infants born prior to week 34 and twins seldom are born after week 38, the variability is actually not that large relative to the general population). We have used chronological age in line with our previous publications, and because age outside the womb (duration of visual experience) is likely to be (more) relevant to the measures in this paper. We have added a sensitivity analysis with corrected age and this is reported in Supplementary Methods 8. The results and conclusions remained the same for the twin and genetic results; for the longitudinal phenotypic associations with developmental outcomes, the statistically significant positive association between preference for the face and receptive vocabulary (comprehension in the CDI) at 14 months was no longer significant, but this was marginal, $p = 0.034$, threshold = 0.03.

Preregistration

This was a pre-registered project and the authors make note of deviations from the pre-registration and report registered analyses in the Supplement.

Appropriate use of statistics and treatment of uncertainties (As noted above):

Given the goal of “experimentally dissociate looking behavior to social (faces) versus non-social objects” and quantifying genetic contributions to both, I am unclear as to why the authors did not conduct symmetric analyses examining first look and sustained attention to faces as well as objects. This approach, followed by measures of the heritability for each, would strengthen the design of this study, better validate the cross-phenotype correlations, and potentially provide stronger evidence for genetic contributions to social but not non-social viewing. What was the rationale for instead simply quantifying numbers of objects attended to?

RESPONSE #7. Please see Response #1 above. Custom

code

R code is available on OSF.

Conclusions

- I find it difficult to interpret results from the non-social object findings given that the

analyses were not comparable to those of the faces

RESPONSE #8. Indeed, the non-social object findings are not at all symmetrical (see Response #1), they are simply a different measure of how infants explore (social and non-social objects). We have gone through the text and revised it in order to minimise the risk that it can be perceived as a purely non-social measure.

- Should “face” be included in the sentence on line 206 p. 16?

RESPONSE #9. This sentence has been clarified.

- The logic of the sentence on p. 17 lines 230-232 is unclear to me. What were the authors’ initial hypotheses (and rationale) regarding this association?

RESPONSE #10. Thank you for pointing out this. We had no directional hypothesis in the preregistration for this association; hence we have chosen to simply remove this sentence from the Discussion.

- The Discussion would be strengthened by contextualizing the findings in relation to the authors’ initial hypotheses.

RESPONSE #11. Thank you for this suggestion. We have now done this, following the hypotheses and open questions in our pre registration.

- The authors should discuss how they interpret the fact that face preference was associated with later receptive but not expressive language.

RESPONSE #12: thanks for this point, we have now added interpretations to this in Discussion (lines 222-225). Please see also our rationale for selecting the two different versions of the scale for the two timepoints (in the Methods section, see also RESPONSE #21 to reviewer #2).

- The authors should acknowledge the limitations of parent-report outcomes **RESPONSE #13:** Thanks for this point, we have added this to the discussion/limitations section (line 237-238).

Suggested improvements

- Rephrase lines 185-187 for clarity

RESPONSE #14. This sentence has been clarified (lines 170-173). Clarity & context

- The abstract should better reflect full extent of the findings from this study, inclusive of null findings. For instance, from the abstract, it is unclear what non-social objects were studied or the associated finding –these should be included as they were core aims of the study.

RESPONSE #15. In line with what we have said above, we see the Object exploration measure as an interesting complementary measure about individual differences in how infants explore social and non-social objects (their tendency to look around quickly vs look at fewer objects for longer time), but this is a separate and independent measure vis a vis the two face looking measures (first looks, % looking). Hence, we do not think it is needed for completeness to include this in the abstract.

Additionally, the null associations with polygenic risk scores should also be mentioned as well as the lack of associations with self-regulation or social communication.

RESPONSE #16. We agree that reporting of null findings is important, but following Editor comments we refrained from interpretation of statistically non-significant results.

- It is unclear what the authors' hypotheses were for the study –the manuscript would be strengthened by presentation of the authors' hypotheses.

RESPONSE #17. Thank you for this good point, we have now included our hypotheses/expectations in the Intro more clearly, following our preregistration.

Reviewer #2:

Remarks to the Author:

Key results

Thank you for the opportunity to read and review the manuscript titled “Infants’ looking preferences for social versus non-social objects reflect genetic variation and are linked to later language development”. This study aims to assess the extent to which individual differences in social attention to faces reflect genetic and environmental factors. The study also aims to understand associations between attention to faces and later developmental skills such as social communication, language, and self-regulation.

Overall, results indicated that orienting to faces as well as sustained attention to faces are heritable. These findings are unique and intriguing in that they consider the impact of genetics on an infant’s visual environment. The findings from this manuscript also indicate that visual preference for faces is positively associated with downstream receptive vocabulary.

Validity

- It is perhaps important to discuss the ecological validity of the stimuli used. This research used a static array of images that have limited ecological validity as a social attentional measure. For instance, the gaze to faces can be impacted by the social content of face (see review by Hessels, 2020).

RESPONSE #18. Thanks for this point, we agree, and have added this point to the limitations paragraph, together with a citation to the mentioned reference (lines 238-239).

Significance

- This research is unique in that it considers the impact of genetic and environmental influences on attention to faces in a large sample of infants.

References

- The reference list is comprehensive and adequate for the topic.

Your expertise

- The reviewer are experts in language, social communication, and autism spectrum disorder. I do not have expertise in genetics and cannot comment on: the use of polygenic scores for autism/ ADHD, nor the justification for using proportion first looks to the face vs latency based on the unmet twin assumption

Providing constructive feedback Data and methodology

- In the methods section, under gaze quality measures, did the authors collect accuracy data? This might be important to considering that eye tracking accuracy is typically poor in infants, and this might impact AOI-based detection of gaze location (Dalrymple et al., 2018).

RESPONSE #19: Please see **RESPONSE #5** to Reviewer #1.

- In the methods section, line 369, the authors state that they used the total score from the CSBS ITC. Was the standard score used here?

RESPONSE #20: The total raw score was used. While the standard score (which is computed following age-based norms) can be used to compare individual scores to other similarly-aged children, we believe we do not need to use them because we are only interested in individual differences within our sample. We did use age (in days) as a

covariate in the longitudinal models, hence our effects consider the age of the infant in the same way. In fact, it can be argued that our approach is better than using normed scores, as normed scores often are adjusted by e.g., month, meaning that the age correction will be less continuous/correct than correcting for each child's actual age in days.

- The rationale to investigate CDI receptive vocab at 14 months and CDI expressive vocab at 24 months was not developed.

RESPONSE #21: Thanks for pointing this out. The reason was that at 14 months there is a substantial floor effect for production. However, the CDI reliably measures individual differences in language production at 24 months and does not include comprehension/receptive vocab because it is judged to be typically too large for parents to report on. We have added this info to the Methods section (lines 351-357).

- Can the authors confirm that Table 1 presents data in the form of % and not proportions?

RESPONSE #22: This has been fixed now.

- The AOIs are quite large. Did the authors confirm that there is sufficient distance in between the AOIs? A common metric is to ensure that 1 degree visual angle.

RESPONSE #23: The stimuli was optimised for infant eye tracking, which can include more spatial errors than data from older participants. Specifically, the stimuli are 'sparse', i.e., there is quite a lot of space between the objects in the array (>>1 degree visual angle). We would like to argue that it is the distance between the stimuli that are most important, not the distance between the AOIs. We opted for creating relatively large AOIs covering these spatially separated objects, to increase the chances that even infants with less accurate data would be included. Notably, given the distance between the objects, the likelihood that an infant would look towards the border of the AOI is very low. The AOIs were drawn after having inspected gaze heatmaps/replays which supported this approach.

Analytical approach

- Results section, Table 3: In this table, the 95% CI for the correlations for MZ and DZ twins overlap. However, the authors conclude that correlations were lower for DZ twins. There seems to be considerable variation in the correlations, which may impact drawing the conclusion that the groups correlations were significantly different. The authors should directly test the differences in a single model if they want to make such conclusions.

RESPONSE #24. Thanks for this important point. Indeed, we cannot say based on the ICCs alone whether the difference (MZ-DZ) is significant. We have now revised the text to make clear that in this paragraph we talk about descriptive patterns (MZ vs DZ) that are in line with, or not in line with, genetic influences (lines 102-110). We want to emphasise that our main results and conclusion do not build on these descriptive patterns, but are based on the full twin models, which are the formal way to test the significance of genetic, shared environment, and unique environment effects. See also response to **Response #37** to Reviewer #3.

- Many twins are born prematurely. Do the authors use a chronological age corrected for gestational age?

RESPONSE #25: Please see **RESPONSE #6** to Reviewer #1.

- The type of FDR correction that used should be specified in the main manuscript, along with how the correction was applied (across all analyses, across analyses using the same DV, ect)

RESPONSE #26. This has now been specified in methods “Analysis plans” section (lines 395-397).

- Results from null hypothesis testing are either significant or they are not significant. Referencing results as “marginal” (line 138) is not appropriate.

RESPONSE #27. Following yours and the Editor comments we have removed interpretations of statistically non-significant results.

- I commend the authors for reporting null results and for assessing group differences between infants that were excluded versus those that were included.

Clarity and context

- In the introduction, the authors note that attending to social vs nonsocial is not “straightforward”. Can the author elaborate on why they think the process is not straightforward?

RESPONSE #28. This section is slightly altered, and we do no longer use this formulation.

- The authors claim to look at ASD and ADHD “outcomes”. This seems like an odd phrase given that they only looked at social communication (their autism “outcome”) at 14 months of age. ASD is not often diagnosed until 2 years of age or much older, so claiming to measure

an outcome at 14 months seems inappropriate. The authors should also be mindful of falling into trap that thinking measuring autism traits in the general population is a way to study autism (see Sasson and Bottema-Beutal, 2021, <https://doi.org/10.1177/13623613211058515>).

RESPONSE #29. Thank you for this point, we have changed to 'autism and ADHD related traits'.

- In the discussion section, (line 241) the authors refer to a report of atypical engagement with social scenes in infants later diagnosed autistic. There has been considerable variability in the social attention literature for autistic infants (see Guillion et al., 2014 for a review on this topic). The authors may consider including this in the discussion. I would caution against a statement based on just one report on this topic given the variability of findings.

RESPONSE #30. Thanks for this important point; however due to the Editor's point of removing all discussion of non-significant results this part is now entirely removed.

- I wonder if the authors could edit the same sentence to improve clarity (discussion, line 240). The sentence starts off discussing the association between face preference and later social communication and language and report that this is in line with reports in autistic infants. Correct me if I am wrong, but the reference here, Chawarska (2013) did not measure associations to later language. Young et al. (2009) might be a useful reference here. Additionally, the authors state "which are in line with reports.." but provide a singular reference.

RESPONSE #31. As noted in the previous response, these sentences have been removed.

Suggested (minor) improvements

- Throughout the manuscript, where the author refers to statistical values, they need to be italicized. For instance "p-values", "R²" etc.

RESPONSE #32: Thank you, we have now edited these.

- Results section, paragraph 1: Can the authors include effect sizes for the t values? Similar comment for Table 5, results section.

RESPONSE #33. Effect sizes for the t values (Cohen's D) have now been added to the manuscript. For Table 5, we believe that the beta values presented can be informative (since all variables were standardized before the analysis) and that adding effect sizes could reduce

the readability of the table. We present the effect size (R^2) of the significant finding.

Reviewer #3:

Remarks to the Author:

Review Portugal et al, Nat Hum Behav 19209_0

I appreciate the chance to read and review this paper analyzing genetic and other influences on infants' looking preferences. The authors have a very large study group of same-sex twins, and have both genomic data and eye-tracking assay measures. As one would expect, they find that infant social visual preferences have measurable heritability. These authors have used many of the heritability techniques in this paper previously, most notably in a 2021 JCPP paper looking at the same study sample but different scientific question. It is not surprising to me that there would be insignificant correlations between face orienting/preference and any of the polygenic scores calculated by the authors, even that for ASD, given the state of that research area.

Strengths:

Study has eye tracking data from one timepoint and parent-report questionnaire from three timepoints in the same cohort and have DZ and MZ twin pairs, along with genomic data.

Authors have analyzed influence of genetics vs environment on face orienting and face preference. This is a fascinating question.

Size of study group is commendably larger than previous studies in this arena.

Problems:

Context statements are not supported by the data. Lines 35-38: this is not the first study to show that individual perceptual choices are heritable. At a minimum, this must be reworded to reflect the novelty actually reported in the paper.

RESPONSE #34. Thanks for this point, we have now reformulated this part accordingly.

Similarly, lines 50-54 are just not correct. Reference #9, cited in this very paper, exactly looked at "to what extent visual preference for faces versus non-social information in infancy reflects genetic variation in the population" as claimed in lines 51-52 here (see extended data figure 6,

<https://pubmed.ncbi.nlm.nih.gov/28700580/#&gid=article-figures&pid=extended-data-figure-6-uid-5>). In fact, that reference had MZ twins, DZ twins, and other pairs, which is a more thorough design than this study for asking the question about G vs E.

RESPONSE #35. We have now reformulated this part accordingly.

Insufficient details in the references, lines 356-364, on calculating polygenic risk scores. Those calculations involve multiple filters and decision points, and what's reported now is not sufficient to aid replication or understanding in context. I see this is a quite similar methods paragraph to a previous paper by the authors, <https://acamh.onlinelibrary.wiley.com/doi/10.1111/jcpp.13564>, but with even fewer details. (A more minor point: unfortunately, for most conditions, 2019 papers are not now the most recent/up to date GWAS. I understand the authors mean at the time when the calculations were being done, and they may want to more explicitly state that.)

RESPONSE #36. We added now more details on QC, imputation and genetic ancestry to Supplementary Methods 5 (the PRS-CS method does not require any threshold adjustment). We also changed the wording re “the most recent/up to date GWAS”.

I do not understand the choice made by the authors in table 3, to say that there are genetic influences on face orienting with values of 0.20 and 0.05 but NOT on object orienting, with values of 0.05 and 0.15. The range on the confidence intervals obviously overlaps there. I am definitely not a statistician so others may understand this choice, but I would suggest some more explanation at least, as the decision is then to not show any more analyses of object orienting.

RESPONSE #37. Thanks for pointing out the need to clarify the rationale more thoroughly. While confidence intervals may overlap for specific point estimates, the direction of MZ-DZ difference is different for the 3 phenotypes. Most notably, the pattern seen for object exploration is not suggestive of a genetic effect (with MZ being descriptively *lower* than DZ). Considering this, together with the non-significant ICCs for both MZ and DZ for this measure, we chose to exclude the object exploration measure from the multivariate analysis. We have now updated this section (lines 102-110), hopefully it is now clear how these decisions were made. See also **Response #24** to Reviewer #2.

Minor corrections/points:

Line 139 is not a complete sentence: “whereas the shared genetic influences were – Table 4.”

RESPONSE #38. This is fixed now.

More description of the models/tests used in lines 119-122 would be helpful for non- specialists.

RESPONSE #39. We have now added more details on twin-model fitting to the **introduction** of the manuscript (lines 82-94).

Line 211: please forgive me if I missed it, but I did not see power calculations included. Is this something the authors could provide, to support statements of being underpowered? I would consider this optional but nice to have.

RESPONSE #40: We ran a general power analysis prior to data collection, in which we assumed 225 pairs would have valid data, and 50% MZ twins. A study of this size has nearly 97% power to detect a heritability of 40% and a shared environmental effect of 40% for a phenotype. We will have nearly 88% power to detect a significant genetic contribution to a correlation between two measures, assuming that the heritabilities of the two variables are 40%, that the shared environment explains 40% of the variation in the two variables, and that the phenotypic correlation is $r = 0.40$ and to the same degree mediated by genetic and shared environmental effects. We have now included this information in the Supplementary Methods 6.

The 2021 Portugal et al paper (congratulations to the authors!) in JCPP used such similar methods that I am a little surprised not to see it cited here.

RESPONSE #41. It was not cited because it studied a very different phenotype (pupillary light reflex). We now provide a citation in the methods (line 275).

Questions more out of curiosity:

Given the time which has elapsed since the study was conducted, it is likely that some of the participants would have received ASD/ADHD diagnoses at this point. For a reported prevalence of ~1% in Sweden overall and 3.66% in an immigrant group in Sweden (Linnsand et al JADD 2021), the authors should see several twin pairs at least who have already been diagnosed. In fact this could have been reported at the 36-month questionnaire, of course. Can they quickly go into the study records and find out if this is the case? It would be fascinating to how genomic or eye tracking markers corresponded to later diagnosis of twin pairs.

RESPONSE #42: We agree this is interesting, but this would require a major effort and/or resources we do not currently have (e.g., it would require contacting the families again or requesting access to registry data, a new ethical application).

I see that in reference 18, the ethnicity of this sample is described as “Visual inspection of the results of the PCA confirmed that the sample was largely homogenous in terms of Swedish/European ancestry, with only a minority of mixed genetic ancestry or non-European genetic ancestry.” To what extent would the authors expect to see these influences in their GxE calculations, given that E might be more noticeable in ASD studies than in those for other neurodevelopmental conditions? Have they accounted for this / can it be accounted for?

RESPONSE #43: In this study, we used two methods, twin model-fitting to estimate twin heritability, and polygenic score association analyses. Twin heritability is, by definition, specific to the population in which it is estimated, and it is not routine to remove individuals based on specific ancestries. Whether ancestry influenced our twin results cannot be addressed in our study given the sample size and the small number of non-European participants, however, we speculate that it is unlikely that heritability estimates were different in this basic phenotype this early in infancy.

For polygenic score analyses, it is often the case that samples with a single homogenous ancestry are studied and as such outliers as defined based on ancestry are usually removed. We re-ran our polygenic score analyses with the outliers for ancestry removed (only one pair of twins) and the results did not change (they are still null).

Our study did not specifically model GxE (i.e. gene-environment interaction) so we will not speculate on these processes which are beyond the scope of this paper.

Decision Letter, first revision:

4th July 2023

Dear Dr. Portugal,

Thank you for your patience as we've prepared the guidelines for final submission of your Nature Human Behaviour manuscript, "Infants' looking preferences for social versus non-social objects reflect genetic variation" (NATHUMBEHAV-22020353A). Please carefully follow the step-by-step instructions provided in the attached file, and add a response in each row of the table to indicate the changes that you have made. Please also address the additional marked-up edits we have proposed within the article file and reporting summary. Ensuring that each point is addressed will help to ensure that your revised manuscript can be swiftly handed over to our production team.

We would hope to receive your revised paper, with all of the requested files and forms within two-three weeks. Please get in contact with us if you anticipate delays.

Nature Human Behaviour offers a Transparent Peer Review option for new original research manuscripts submitted after December 1st, 2019. As part of this initiative, we encourage our authors to support increased transparency into the peer review process by agreeing to have the reviewer comments, author rebuttal letters, and editorial decision letters published as a Supplementary item. When you submit your final files please clearly state in your cover letter whether or not you would like to participate in this initiative. Please note that failure to state your preference will result in delays in accepting your manuscript for publication.

In recognition of the time and expertise our reviewers provide to Nature Human Behaviour's editorial process, we would like to formally acknowledge their contribution to the external peer review of your manuscript entitled "Infants' looking preferences for social versus non-social objects reflect genetic variation". For those reviewers who give their assent, we will be publishing their names alongside the published article.

Cover suggestions

As you prepare your final files we encourage you to consider whether you have any images or illustrations that may be appropriate for use on the cover of Nature Human Behaviour.

Please submit your suggestions, clearly labeled, along with your final files. We'll be in touch if more

information is needed.

ORCID

Non-corresponding authors do not have to link their ORCIDs but are encouraged to do so. Please note that it will not be possible to add/modify ORCIDs at proof. Thus, please let your co-authors know that if they wish to have their ORCID added to the paper they must follow the procedure described in the following link prior to acceptance:

Nature Human Behaviour has now transitioned to a unified Rights Collection system which will allow our Author Services team to quickly and easily collect the rights and permissions required to publish your work. Approximately 10 days after your paper is formally accepted, you will receive an email in providing you with a link to complete the grant of rights. If your paper is eligible for Open Access, our Author Services team will also be in touch regarding any additional information that may be required to arrange payment for your article. Please note that you will not receive your proofs until the publishing agreement has been received through our system.

Please note that *Nature Human Behaviour* is a Transformative Journal (TJ). Authors may publish their research with us through the traditional subscription access route or make their paper immediately open access through payment of an article-processing charge (APC). Authors will not be required to make a final decision about access to their article until it has been accepted. Find out more about Transformative Journals

[REDACTED]

Best regards,
Alex McKay
Editorial Assistant
Nature Human Behaviour

On behalf of

Charlotte Payne

Charlotte Payne, PhD
Senior Editor
Nature Human Behaviour

Reviewer #1:

Remarks to the Author:

I believe that the authors have sufficiently responded to reviewer concerns and I applaud their efforts on this manuscript.

Reviewer #2:

Remarks to the Author:

The authors have been largely responsive to previous comments by this reviewer. Below are the few outstanding items that I think should be addressed.

1. Regarding the reporting of the CSBS ITC- thank you for the clarification that total raw scores were used, but this should be reported in the manuscript. This could be done by simply adding “raw” to page 15 line 345.
2. Regarding the issue of correcting for gestational age- the authors provide no justification for their statement that “age outside the womb is more relevant to the measures of the paper”. This seems counter to paper reporting that gestational age can have significant impacts on receptive language (Snijders et al., 2020). While I appreciate that it can be frustrating to find results that are just past a significance threshold, with null hypothesis testing there is either significant results or not significant results, there is no “marginal”. In the discussion the authors should be sure to include the not significant

results between face preference and receptive vocabulary when controlling for gestational age when talking about the findings in the main paper.

3. Related- page 7 line 137 still uses the term “marginal” when talking about not significant results. This should be removed.

4. Regarding “descriptive patterns” in the DZ and MZ correlations- the author’s response was not satisfactory and lines 102- 106 on page are problematic. Either empirically test the difference in the correlations, or just rely on the univariate analysis to make the point. As written now, the text (in my opinion) is misleading.

Reviewer #3:

Remarks to the Author:

Second review for NATHUMBEHAV-22020353

First, congratulations on your new baby! That is the best possible reason for a delay in resubmission, and I hope everyone is doing well.

I appreciate the clarifications and rewordings on novelty concerns and PRS calculations. I think the rewording in lines 102-110 is better but would not support the repeated use of the term “descriptively” to report non-significant conclusions there – I will leave that to the editor’s discretion.

Thank you for including more methods details and the power calculations, all of which make this a stronger submission. I would put in at least a sentence with the results of the power calculation at lines 196-7 instead of referring the readers to SI for all of it, but again the editor can make that call here.

Author Rebuttal, first revision:

Reviewer #1 (Remarks to the Author):

I believe that the authors have sufficiently responded to reviewer concerns and I applaud their efforts on this manuscript.

Reviewer #2 (Remarks to the Author):

The authors have been largely responsive to previous comments by this reviewer. Below are

the few outstanding items that I think should be addressed.

1. Regarding the reporting of the CSBS ITC- thank you for the clarification that total raw scores were used, but this should be reported in the manuscript. This could be done by simply adding “raw” to page 15 line 345.

RESPONSE #1: This has been added now.

2. Regarding the issue of correcting for gestational age- the authors provide no justification for their statement that “age outside the womb is more relevant to the measures of the paper”. This seems counter to paper reporting that gestational age can have significant impacts on receptive language (Snijders et al., 2020). While I appreciate that it can be frustrating to find results that are just past a significance threshold, with null hypothesis testing there is either significant results or not significant results, there is no “marginal”. In the discussion the authors should be sure to include the not significant results between face preference and receptive vocabulary when controlling for gestational age when talking about the findings in the main paper.

RESPONSE #2: We have included this point now in the following way:

“However, only the association between face preference and receptive vocabulary at 14 months was significant applying stringent statistical criteria (Table 5, although it was not significant when controlling for gestational age instead of chronological age).”

3. Related- page 7 line 137 still uses the term “marginal” when talking about not significant results. This should be removed.

RESPONSE #3: We have now removed reference to “marginal”.

4. Regarding “descriptive patterns” in the DZ and MZ correlations- the author’s response was not satisfactory and lines 102- 106 on page are problematic. Either empirically test the difference in the correlations, or just rely on the univariate analysis to make the point. As written now, the text (in my opinion) is misleading.

RESPONSE #4: In response to the reviewer and the editor comments, we have reworded lines

102- 106 so that we rely only on the univariate model results to make our point.

Reviewer #3 (Remarks to the Author):

Second review for NATHUMBEHAV-22020353

First, congratulations on your new baby! That is the best possible reason for a delay in resubmission, and I hope everyone is doing well.

I appreciate the clarifications and rewordings on novelty concerns and PRS calculations. I think the rewording in lines 102-110 is better but would not support the repeated use of the term “descriptively” to report non-significant conclusions there – I will leave that to the editor’s discretion.

RESPONSE #5: See RESPONSE #4 above.

Thank you for including more methods details and the power calculations, all of which make this a stronger submission. I would put in at least a sentence with the results of the power calculation at lines 196-7 instead of referring the readers to SI for all of it, but again the editor can make that call here.

RESPONSE #6: We believe the information about power calculations to be appropriately placed in the SI, but we are happy to move it if the editor disagrees.

Final Decision Letter:

Dear Dr Portugal,

We are pleased to inform you that your Article "Infants' looking preferences for social versus non-social objects reflect genetic variation", has now been accepted for publication in Nature Human Behaviour.

Please note that *Nature Human Behaviour* is a Transformative Journal (TJ). Authors may publish their research with us through the traditional subscription access route or make their paper immediately open access through payment of an article-processing charge (APC). Authors will not be required to make a

final decision about access to their article until it has been accepted. Find out more about Transformative Journals

With best regards,

Charlotte Payne

Charlotte Payne, PhD
Senior Editor
Nature Human Behaviour